# LC-MS/MS Method for the Quantification of PARP Inhibitors Olaparib, Rucaparib and Niraparib in Human Plasma and Dried Blood Spot: Development, Validation and Clinical Validation for Therapeutic Drug Monitoring

**DOI:** 10.3390/pharmaceutics15051524

**Published:** 2023-05-18

**Authors:** Giovanni Canil, Marco Orleni, Bianca Posocco, Sara Gagno, Alessia Bignucolo, Marcella Montico, Rossana Roncato, Serena Corsetti, Michele Bartoletti, Giuseppe Toffoli

**Affiliations:** 1Experimental and Clinical Pharmacology Unit, CRO Aviano, National Cancer Institute, IRCCS, 33081 Aviano, Italy; giovanni.canil@cro.it (G.C.); marco.orleni@cro.it (M.O.); sgagno@cro.it (S.G.); alessia.bignucolo@cro.it (A.B.); rroncato@cro.it (R.R.); gtoffoli@cro.it (G.T.); 2Doctoral School in Pharmacological Sciences, University of Padua, 35131 Padova, Italy; 3Clinical Trial Office, CRO Aviano, National Cancer Institute, IRCSS, 33081 Aviano, Italy; marcella.montico@cro.it; 4Department of Medical Oncology, CRO Aviano, National Cancer Institute, IRCCS, 33081 Aviano, Italy; serena.corsetti@cro.it (S.C.); michele.bartoletti@cro.it (M.B.)

**Keywords:** LC-MS/MS, PARP inhibitors, olaparib, rucaparib, niraparib, human plasma, dried blood spot, therapeutic drug monitoring

## Abstract

Poly (ADP-ribose) polymerase inhibitors (PARPis) are becoming increasingly meaningful in oncology, and their therapeutic drug monitoring (TDM) might be beneficial for patients. Several bioanalytical methods have been reported for PARPis quantification in human plasma, but advantages might be obtained using dried blood spot (DBS) as a sampling technique. Our aim was to develop and validate a liquid chromatography-tandem mass spectrometric (LC-MS/MS) method for olaparib, rucaparib, and niraparib quantification in both human plasma and DBS matrices. Additionally, we aimed to assess the correlation between the drug concentrations measured in these two matrices. DBS from patients was obtained using Hemaxis DB10 for volumetric sampling. Analytes were separated on a Cortecs-T3 column and detected with electrospray ionization (ESI)-MS in positive ionization mode. Validation was performed according to the latest regulatory guidelines, in the range (ng/mL) 140–7000 for olaparib, 100–5000 for rucaparib, and 60–3000 for niraparib, within the hematocrit (Hct) range 29–45%. The Passing–Bablok and Bland–Altman statistical analyses revealed a strong correlation between plasma and DBS for olaparib and niraparib. However, due to the limited amount of data, it was challenging to establish a robust regression analysis for rucaparib. To ensure a more reliable assessment, additional samples are required. The DBS-to-plasma ratio was used as a conversion factor (CF) without considering any patient-related hematological parameters. These results provide a solid basis for the feasibility of PARPis TDM using both plasma and DBS matrices.

## 1. Introduction

Therapeutic drug monitoring (TDM) is an emerging approach that aims to reach optimal drug dosing in patients to ensure the best clinical outcome. The standard procedure for TDM is to measure the drug concentration in plasma at a specific and pharmacokinetically meaningful time point. Fundamental parameters that allow the application of TDM in clinical practice are a straightforward relationship between exposure and efficacy/safety [1,2] and a defined exposure value used as a target value. In such cases, knowing the drug concentration during treatment can help clinicians make the best decision for the patient’s clinical outcome. This is of great importance for recent modalities of drug treatment based on daily oral administration, where the minimum drug concentration before the next administration (C_trough_) correlates with treatment efficacy and/or toxicity [1,2].

Poly (ADP-ribose) polymerase inhibitors (PARPis) are a relatively new class of targeted drugs in oncology and are considered good candidates for TDM [1,2,3]. To date, olaparib, rucaparib, and niraparib are among the most commonly used PARPis in the clinic. They are used for maintenance therapy of platinum-sensitive advanced or recurrent ovarian cancer and HER2-negative locally advanced or metastatic breast cancer [4,5,6,7,8,9,10,11]. In addition to these indications, they are also being tested in several clinical trials in other cancers as monotherapy and combination therapy [12,13]. Their use is expected to increase in the near future, and their routine quantification by TDM could play a major role in clinical practice [14,15,16,17]. PARPis meet most of the criteria for drug suitability for TDM, as indicated by Groenland et al. [1]. In particular, they are prescribed as long-term therapy, taken daily for at least two years or until disease progression or unacceptable toxicity [18,19,20,21]. In addition, their dose can be easily adjusted to individual patient needs because PARPis are formulated as fixed-dose capsules or tablets [8,9,10,11,22]. Moreover, there are no measurable biomarkers to evaluate the effects of the drug [4,5,6,7,23], and validated bioanalytical methods have been published to quantify them, as reviewed recently [24]. Finally, PARPis exhibit considerable variability in pharmacokinetic (PK) exposure, as shown by relevant studies on olaparib [3,25], niraparib [26], and rucaparib [6,27]. Furthermore, an association between exposure and toxicity and/or efficacy has been reported for all PARPis [18,19,20,28,29].

Currently, the TDM of PARPis is still exploratory, as plasma C_trough_ target values have not yet been defined. However, Verheijen et al. reported that in cases where there is no evidence-based target exposure, the average exposure of the approved effective dose can serve as a proxy [30]. The preliminary TDM target for olaparib is 1290 (CV% = 76%) ng/mL [25], for niraparib is 649 ± 135 ng/mL [26], and for rucaparib is 1754 ± 805 ng/mL [6]. Routine clinical analyses for TDM are usually performed in human plasma. Several methods for quantifying PARPis in plasma using liquid chromatography-tandem mass spectrometry (LC-MS/MS) have been reported in the literature. These methods involve chromatography, mass spectrometry, and validation of analytical characteristics. A recent review provides a comprehensive overview of these methods, including their validation and other analytical characteristics [24]. Among them, only the method of Bruin et al. [16] allowed the analysis of all analytes relevant to the present work. Nevertheless, alternative sampling methods for TDM have recently been investigated and reported. One such method is the collection of dried blood spot (DBS) on paper using a simple fingerprick technique [31,32,33]. DBS has some obvious advantages compared to venipuncture. For instance, it is a less invasive, more cost-effective method that does not require specialized personnel for sample collection. With DBS, patients can even be trained to collect their own samples at home, especially when it is difficult to collect samples at pharmacokinetically appropriate times, such as C_trough_. This is particularly important for niraparib, which is usually taken before bedtime to reduce side effects such as nausea.

As pharmacological research on PARPis is exploring exposure–efficacy/safety relationships to potentially define a concentration threshold for future use of TDM, we believe that DBS will increase the dissemination of these studies. Methods for quantifying PARPis in DBS are, to the best of our knowledge, not yet available in the literature.

For these reasons, we aimed to develop and validate a simple and robust LC-MS/MS quantification method for olaparib, rucaparib, and niraparib (Figure 1) suitable for routine TDM in both plasma and DBS matrices. We developed and validated a new method for quantifying PARPis in plasma samples due to concerns regarding the validation protocol of the previously mentioned method [16]. The previous protocol, which was considered suitable for TDM [34], deviated in some aspects from the guidelines provided by the European Medicines Agency (EMA) and Food and Drug Administration (FDA). Moreover, only one transition was selected for the detection of each analyte, while each analyte and the corresponding internal standard (IS) have the same *m*/*z* value of the product ion. The plasma-based method was used as a reference for validating the quantification of PARPis in the DBS matrix, which was the ultimate goal of this work.

## 2. Materials and Methods

### 2.1. Materials Used

Olaparib, rucaparib, and niraparib reference standards and [^2^H_8_]-Olaparib, [^13^C,^2^H_3_]-Rucaparib, [^13^C_6_]-Niraparib hydrochloride used as ISs, were purchased from Alsachim (Grand Est, France). Formic acid (puriss. 98–100%) and acetonitrile (hypergrade for LC-MS) were purchased from Merck (Darmstadt, Germany). Ammonium formate (for mass spectrometry ≥ 99%) and ammonium hydroxide (28%, purity > 99.9% trace metal bases) were obtained from Sigma Aldrich (St. Louis, MO, USA). Methanol (LC-MS grade) was purchased from Carlo Erba (Milano, Italy), and MilliQ water was obtained using an in-house system IQ 7000 from Merck (Darmstadt, Germany). The Whatman 903 paper was purchased from GE Healthcare (Westborough, MA, USA). Finger prick blood samples were collected using Accu-Check Safe-T-Pro Plus lancets (1.8 mm penetration depth, 0.63 mm gauge) from Roche Diagnostics (Mannheim, Germany), and the Hemaxis DB10 whole blood collection device from DBS System SA (Gland, Switzerland) for volumetric sampling (10 µL). K_2_EDTA blank human plasma and whole blood were supplied by the transfusion unit of our institute. A set of Gilson Microman-E Pipettes (Middleton, WI, USA) was used to handle organic solvents and plasma/blood samples.

### 2.2. Stock and Working Solutions

Stock solutions of all reference standards (in duplicate) and stable isotope-labeled ISs were prepared in methanol and stored at −80 °C. The two separate stock solutions of each reference standard were diluted with methanol to prepare the working solutions (WSs) of the upper limit of quantification (ULOQ) and quality control (QC) H. This means that the calibrator and QC have the highest concentration, respectively. The calibrators and QCs were obtained by diluting ULOQ and QCH with methanol and stored them at −80 °C. The IS WSs for the method in human plasma were prepared by diluting the IS stock solutions with acetonitrile. Similarly, the IS WS for the method in DBS was prepared by diluting the IS stock solutions with methanol. Both IS WSs were stored at −20 °C. Finally, a WS exceeding the ULOQ concentration was prepared in methanol to test dilution integrity in plasma. The concentrations of the stock solutions and WSs can be found in Appendix B.

### 2.3. Human Plasma Sample Preparation

Plasma calibrators and QC samples were prepared by spiking 10 µL of the corresponding WS into 190 µL of human plasma. The mixture was vortex-mixed for 10 s, followed by centrifugation. Subsequently, 50 µL aliquots were stored at −80 °C until analysis. The final plasma concentrations are listed in Table 1. Plasma from patients treated with olaparib, rucaparib, and niraparib was obtained by centrifugation of their whole blood at 2608 relative centrifugal force (RCF) for 10 min at 4 °C. The resulting plasma was then stored at −80 °C until analysis. For the extraction procedure, the plasma samples were thawed at room temperature and vortexed for 10 s. Then, 250 µL of the proper IS WS was added to 50 µL of plasma and vortexed for 10 s. The mixture was centrifuged at 16,200 RCF for 10 min at 4 °C. After centrifugation, 50 µL of the supernatant was added to 700 µL of mobile phase (MP)-A and vortexed for 10 s. Following additional centrifugation at 16,200 RCF for 10 min at 4 °C, 200 µL of the final extract was transferred to a polypropylene (PP) autosampler vial (Agilent, Santa Clara, CA, USA). The vial was then stored at 15 °C until analysis. For analysis, a 5 µL aliquot of the extract was injected into the system. 

### 2.4. DBS Sample Preparation

DBS calibrators and QC samples were prepared using a 0.5 mL polypropylene (PP) tube. Initially, 10 µL of the corresponding WS was spiked into 190 µL of human whole blood. The Hct of the blood was adjusted to 36% using the plasma removal and addition procedure described by Koster et al. [35]. It should be noted that after the procedure, the Hct was not verified using a hematology analyzer. Then, the samples were gently mixed and incubated at 37 °C for 30 min in a Vortemp 56 (Illumina, San Diego, CA, USA). During incubation, the samples were stirred at a speed of 300 rpm. After incubation, 10 µL of the mixture was spotted on Whatman 903 paper. The spotted samples were then dried at room temperature for at least 1 h. Subsequently, the dried samples were stored in a Sicco Star desiccator (Bohlender, Grünsfeld, Germany) at a temperature of 20 °C and a humidity level below 35% until analysis. The final blood concentrations of the blood samples are listed in Table 1). Patients’ DBS samples were obtained from capillary blood by pricking their fingers with the Accu-Check lancet and collecting the blood in the Hemaxis DB10 device. The first blood drop was removed and the subsequent drops were allowed to dry open in the Sicco desiccator for one night. Once dried, the DBS samples were sealed and kept in a sealed condition until analysis. The extraction procedure was as follows: DBSs were punched with an 8-mm-diameter manual puncher allowing volumetric sampling (10 µL). The punched paper disks were then transferred to a 2-mL PP tube. Subsequently, 250 µL of the appropriate IS WS was added to the tube. The contents of the tube were mixed using a microplate shaker (VWR, Radnor, TN, USA) for 30 min at 20 °C and a speed setting of 500. Then, 50 µL of the supernatant obtained after centrifugation was added to 200 µL of MP-A and vortex-mixed for 10 s. After spinning, 200 µL of the final extract was transferred to a PP autosampler vial and kept at 15 °C until analysis. The injected volume varied from 10 to 25 µL, depending on the signal obtained. It is important to note that no differences in chromatography were observed during the analysis.

### 2.5. Chromatographic and Mass Spectrometric Conditions

The LC system consisted of a degassing unit (DGU-20A3R), four Nexera XR pumps (LC-20AD), a Nexera XR autosampler (SIL-20AC), a Prominence oven (CTO-20AC), and a controller (CBM-20A), all from Shimadzu (Tokyo, Japan). The instrument used in the study had a dwell volume of about 370 µL. The extra column volume (ECV) was around 25 µL. The determination of ECV was based on established procedures described by most LC manufacturers, where acetone is replaced with uracil for MS detection. The chromatographic column was a Cortecs T3 (75 × 3 mm, 2.7 µm, 120 Å) with a security guard (VanGuard T3, 2.1 × 5 mm, 2.7 µm) from Waters (Milford, KS, USA). Additionally, an in-line filter (0.5 µm) was used to further protect the column. MPs consisted of a mixture of an aqueous buffer (formic acid/ammonium formate 10/10 mM, pH 3.7, MP-A) and methanol (MP-B). Chromatographic separation was performed using a linear gradient (Appendix A), a flow rate of 0.6 mL/min, a column oven at 40 °C, and an autosampler at 15 °C. A Valco valve directed flow from the LC to the MS between 1.50 and 6.00 min. For the remaining runtime, the valve was set to direct the flow to waste. From 0.00 min to 1.50 min and from 6.00 min to 12.00 min, a mixture of water/methanol (80/20 *v*/*v*) was directed to the MS. The needle wash solution consisted of 0.3% ammonium hydroxide in a mixture of methanol/water (80/20 *v*/*v*).

The MS was a 4000 Qtrap triple quadrupole from AB Sciex (Framingham, MA, USA) equipped with an electrospray ionization (ESI) source operating in positive ionization mode; The MS acquisition method utilized multiple reaction monitoring (MRM) modes for targeted analysis of specific analytes. Comprehensive details regarding the MS acquisition method, including specific MRM transitions and optimized parameters, are summarized in Appendix A. Independent solutions containing 200 ng/mL of each molecule in methanol were used to obtain the optimized MRM acquisition mode parameters in both positive and negative ionization modes. The ESI source parameters were optimized by infusing a 200 ng/mL solution of the analytes in methanol along with a flow of MPs (MP-A/MP-B 50/50 *v*/*v*) from LC. Analyst software (version 1.6.3, AB Sciex) was used for data acquisition, and Multiquant (version 2.1, AB Sciex) for quantification.

### 2.6. Validation of the Method in Human Plasma

The validation was performed according to the latest guidelines of the EMA [36] and FDA [37]. In particular, linearity, accuracy (expressed as mean % accuracy), precision (the imprecision was expressed as coefficient of variation—CV%), recovery, matrix effect, selectivity, sensitivity (lower limit of quantification—LLOQ), dilution integrity, and stability were evaluated. The details of these tests are briefly described below.

#### 2.6.1. Linearity

Linearity was assessed with two different sets of eight calibrators in nine different runs of analysis on nine different days. The heteroscedasticity of calibration values was assessed using an F-test [38]. To select the best weighting factor (w, where w = 1, 1/x, or 1/x^2^) to apply to the linear regression model, the following statistical parameters were calculated: the sum of squares (SS), the correlation coefficient (r), the Akaike’s information criterion (AIC), and the sum of absolute %RE (percent relative error). In addition, the %RE was plotted against nominal concentrations (*C_nom_*) to compare back-calculated concentrations (*C_found_*) computed from the regression equation obtained for each w. The %RE was calculated as follows:$$\%RE=\frac{C_{found}-C_{nom}}{C_{nom}}*100$$

The best w is the one that leads to a narrow horizontal band of randomly distributed %RE around the concentration axis and has the smallest sum of %RE. Additionally, w is selected based on the smallest values of SS and AIC and the r-value closest to 1.

#### 2.6.2. Accuracy and Precision

Accuracy and precision (A & P) were determined in the analysis of five LLOQ, five QCL, five QCM, and five QCH. Within-run A & P were assessed on a single working day, whereas between-run A & P were determined in three different analytical runs on three different days. The measured concentrations had to be within ±15% of the nominal value, ensuring an accuracy range of 85% to 115%). Additionally, the imprecision (coefficient of variation) needed to be ≤15% for most samples. However, for the LLOQ samples, the accuracy range should be between 80% and 120%, with imprecision ≤ 20%. QC concentrations were set as follows: QCL is 2.55 higher than the LLOQ, QCM is about 35% of the range, and QCH is 85% of the ULOQ.

#### 2.6.3. Recovery and Matrix Effect

Recovery was evaluated as the ratio between the analyte signal obtained in the normal extraction and the signal obtained in the post-extraction technique (blank plasma spiked with analyte after extraction). It was assessed using QC samples at L, M, and H concentration levels prepared in quintuplicate.

The matrix effect was evaluated by comparing the peak area obtained with the post-extraction technique with that obtained with the pure solution (in methanol) of the analytes at QCL and QCH concentrations. For the preparation of the “post-extraction” QCs, six different healthy donors (three males and three females) and a plasma with mild hemolysis were used. Samples containing analytes in methanol were prepared in triplicate. In addition, the IS-normalized matrix factor (ISN-MF) was calculated as the ratio between the matrix factor of the analyte and the IS matrix factor. The CV% of ISN-MF should be ≤15%.

#### 2.6.4. Selectivity and Sensitivity

The same matrices were used to evaluate selectivity for the matrix effect (six different donors and a plasma sample with mild hemolysis). The method was considered free of nonspecific interference if the response was <20% of the LLOQ for the analyte and <5% for the IS at the retention times of the analytes and ISs.

The lowest non-zero standard on the calibration curve (Table 1), LLOQ, defines the sensitivity of the method. The LLOQ response should be five times higher than that of the zero samples and have a signal-to-noise ratio of at least ≥5. Sensitivity was evaluated in three different runs using LLOQ samples prepared in quintuplicates as part of the A & P: accuracy had to be between 80–120%, while CV% had to be ≤20%.

#### 2.6.5. Dilution Integrity and Stability

Dilution integrity was evaluated on five samples that exceeded the ULOQ concentration and were diluted with pooled plasma. A dilution factor of five was evaluated to cover all possible concentrations in the patient’s samples. Stability in plasma was assessed on the bench at room temperature, at −80 °C, and after several freeze–thaw cycles. Moreover, the stability of the final extract was tested under autosampler conditions (15 °C, dark). All measured concentrations had to be within ±15% of the nominal value, with CV% ≤ 15%.

### 2.7. Validation of the Method in DBS

The validation was performed following the guidelines by Capiau et al. [39], which adapt many of the EMA and FDA recommendations to make them suitable for the DBS matrix. In this description, only those experiments that differ from the EMA and FDA guidelines and are specifically designed for DBS samples are included. For evaluation of linearity and A & P, please refer to Section 2.6.1 and Section 2.6.2. Selectivity and sensitivity were evaluated according to Section 2.6.4, using six replicates of blank samples from six different donors (three males and three females) to determine selectivity. Dilution was not investigated for the DBS matrix because preliminary data indicated that dilution was not required for the human plasma matrix. In fact, all samples analyzed were within the validated range.

#### 2.7.1. Hematocrit Effect

The Hct range was validated in a single run of A & P with QCs at 4 different levels of concentration (LLOQ, QCL, QCM, and QCH) in triplicate, prepared at Hcts of 29 and 45%. They were read on a calibration curve prepared using blood with an artificial Hct of 36%.

#### 2.7.2. Recovery and Matrix Effect

Recovery and matrix effects were evaluated using the post-extraction technique [40] for human plasma. The evaluation was conducted on samples obtained from six different donors (three males and three females). Additionally, two different artificial Hct levels, namely 29 and 45%, were considered. The analysis was performed in triplicate for low, medium, and high QC. A total of 72 QCs were compared with 72 post-extracted QCs for recovery, and 72 post-extracted QCs were compared with 24 solvent QCs for matrix effect for each analyte.

#### 2.7.3. Carryover and Stability

Carryover was evaluated by injecting a blank sample after ULOQ and punching blank paper after punching the ULOQ calibrator (to assess physical carryover). The final extract stability was evaluated under autosampler conditions (15 °C, dark), while the long-term stability of the DBS was evaluated under desiccator conditions (20 °C, humidity < 35%).

### 2.8. Clinical Validation

All samples were collected from patients participating in a clinical trial (protocol ID: CRO 2022-14, approval date: 12 April 2022) ongoing at the IRCCS National Cancer Institute CRO Aviano (Italy). It was approved by the local ethics committee (Comitato Etico Unico Regionale—CEUR) and conducted in accordance with the principles of the Declaration of Helsinki. Written informed consent was obtained from all participants. Data on sex, Hct, dates of Hct measurement, time of last intake, and compliance to treatment were recorded for all patients enrolled in the study. After the steady state was achieved, venous blood and DBS samples were collected ±30 min apart from each other, except for one withdrawal for olaparib (60 min) and one for niraparib (120 min) due to difficulties. When possible, samples were collected at C_trough_ for future pharmacological studies in the population, which are outside the scope of the present work. Because TDM reference targets are generally reported in the literature as plasma concentrations, the use of DBS samples as an alternative to standard plasma samples requires estimating plasma concentration based on the DBS measurement using a conversion strategy. As reported in the guideline by Capiau et al. [39], a minimum of 40 samples from at least 25 subjects is recommended to obtain a conversion formula. We propose a simple conversion method to obtain the estimated plasma concentration (*EC_pla_*) based on an average conversion factor (*CF*) between the concentration in DBS (*C_DBS_*) and the concentration in plasma (*C_pla_*):$$CF=\frac{C_{DBS}}{C_{pla}}$$

Once the average *CF* was calculated for each analyte, it was applied to obtain the *EC_pla_* based on the DBS measure, as follows:$${EC}_{pla}=\frac{C_{DBS}}{CF}$$

To evaluate the agreement between *EC_pla_* and actual *C_pla_*, different statistical analyses were performed. These included Passing–Bablok regression analysis, Bland–Altman plots, Lin’s concordance correlation coefficient (Lin’s CCC), and Spearman correlation (r_S_). The statistical analyses were performed using STATA 14.2 software (StataCorp, Lakeway Drive, College Station, TX, USA). A *p*-value < 0.05 was considered statistically significant. Passing–Bablok regression analysis was to be linear with an angular coefficient of one and pass through the origin of the axes for best agreement. The Bland–Altman plot illustrated the bias (expressed in ng/mL) between variables; in the best case, the bias is close to 0 ng/mL and its 95% confidence interval (CI) includes zero and is small compared to the measured concentration. Spearman correlation evaluates whether trends are present between the variables considered in the Bland–Altman plot (r_S_ close to zero means no trends are present). Moreover, regulatory agencies such as the EMA and FDA require that the percent difference between *EC_pla_* and *C_pla_* (calculated using the formula below) be within ±20% for at least 67% of the samples analyzed.
$$\%diff=\frac{{EC}_{pla}-C_{pla}}{mean({EC}_{pla},C_{pla})}*100$$

### 2.9. Incurred Sample Reanalysis

A subset of patient samples was analyzed in two separate analytical runs to evaluate the incurred sample reanalysis (ISR) for both plasma and DBS matrices, as required by regulatory agencies [36,37]. ISR provides an additional measure of assay reproducibility. The two analyses can be considered equivalent if the percent difference between the first (*original*) and second (*repeat*) concentrations was within ±20% for at least 67% of the samples analyzed:$$\%diff=\frac{\left(repeat-original\right)}{mean(repeat,original)}*100$$

## 3. Results and Discussion

### 3.1. Development and Optimization of the LC-MS/MS Method

First, the MS compound-dependent parameters were optimized for each analyte and IS. The positive ionization mode was chosen primarily because niraparib was undetectable in the negative mode. For each analyte, one product ion was used for quantification and a second for analyte confirmation, as recommended in several guidelines [41,42]. Quantification was performed using the following transitions: 324 > 293 *m*/*z* for rucaparib, 321 > 304 *m*/*z* for niraparib, 435 > 367 *m*/*z* for olaparib, 328 > 285 *m*/*z* for rucaparib-IS, 327 > 310 *m*/*z* for niraparib-IS, and 443 > 375 *m*/*z* for olaparib-IS. Only one transition was used for the ISs since the product ion of each IS contained the isotopic label within the selected fragment, so there could be no crosstalk between the IS and the co-eluting analyte. The final MS compound-dependent parameters are summarized in Appendix A, while the MS fragmentation spectra of each compound are shown in Appendix A. No elucidation has been made concerning the analyte fragmentation pattern since the experimentally found *m*/*z* transitions were the same as those already described in the literature for Olaparib [16,17,43,44], rucaparib [16,45], and niraparib [16,17,46]. At this stage, the MS source parameters were set to intermediate values to be optimized later.

Subsequently, different chromatographic conditions were investigated and optimized to obtain an efficient and robust chromatographic process. A solution containing 200 ng/mL of each analyte in a mixture of water/methanol 1/1 *v*/*v* was used to test three different columns: Waters XSelect CSH C18 (75 × 3 mm, 3.5 µm), Waters XSelect CSH Phenyl-Hexyl (75 × 3 mm, 3.5 µm), and Waters Cortecs T3 (75 × 3 mm, 2.7 µm). All of these columns were tested using various combinations of MP-B (methanol or acetonitrile) and MP-A (0.1% formic acid or 0.1% ammonium hydroxide in water). A scouting gradient program was employed from 95/5 MP-A/MP-B to 5/95, from 0 to 10 min. The signal intensity of niraparib was always markedly low under basic conditions; therefore, an acidic pH was chosen for chromatography. All the columns tested gave good results in terms of peak shape and separation. However, Cortecs T3 was chosen due to the lower tailing and higher retention of the compounds with methanol and an acidic pH. This column is indeed suitable for the separation of polar molecules under acidic conditions, as it allows some interaction between the analytes and the silanol surface of the particles [47].

The pH of MP-A was fixed at 3.7 by using an aqueous buffer consisting of formic acid and ammonium formate. The buffer concentration used was 10 mM for both formic acid and ammonium formate. This concentration was chosen to minimize tailing of niraparib and rucaparib, ensuring a tailing factor of ≤1.3. The tailing factor was calculated following the method described by Dolan [48]. Lower buffer concentrations resulted in higher tailing, while higher concentrations did not affect the tailing factor.

The chromatographic program was then optimized, resulting in a run of 12 min. Separation was achieved with a flow rate of 0.6 mL/min and a linear gradient from 20% to 90% of MP-B between 0 and 7 min (an increase of 10% B/min). The k* value of the analytes was 4.5, calculated according to Snyder et al. [49]. Figure 2 shows that sufficient separation was obtained for each peak pair. Then, a washing step of 1 min at 90% of MP-B was set to wash the column from the most lipophilic contaminants, while a re-equilibration step of 3.5 min at 20% of MP-B was sufficient to obtain reproducible chromatography. A problem anticipated by van Andel et al. was the potential carryover of niraparib [46]. To address this concern, a basic needle wash solution was employed, consisting of 0.3% ammonium hydroxide in a mixture of methanol/water (80/20 *v*/*v*). The effectiveness of this approach was confirmed by analyzing a blank sample injected after the ULOQ sample. The niraparib and niraparib-IS areas in the blank sample were found to be less than 20% and 5% of the niraparib and niraparib-IS areas in the LLOQ sample, respectively.

Finally, the parameters of the ESI source of the MS were optimized for niraparib, the analyte with the lower concentration range and sensitivity, resulting in similar signal intensities for all analytes. Acquisition dwell times were optimized for each transition to obtain at least 15 points per peak at the LLOQ concentration. The final MS parameters are summarized in Appendix A.

The analytical ranges of each analyte were set to adequately cover the expected steady-state concentrations at trough level in patient samples according to the EMA and FDA documents on the pharmacological properties of PARPis [4,5,6,7,10,37]. Additionally, some relevant publications in the literature [3,14,15,16,17] were considered during the determination of these ranges. As anticipated in the introduction, the average observed C_trough_ for niraparib is 649 ± 135 ng/mL [26]. For rucaparib, it is 1754 ± 805 ng/mL [6], while for olaparib, it is 1290 (CV% = 76%) and 1840 (340–3830) ng/mL [3,25]. Therefore, the range of analytes in both plasma and DBS was set at 60–3000 ng/mL for niraparib, 100–5000 ng/mL for rucaparib, and 140–7000 ng/mL for olaparib.

Subsequently, the sample pretreatment was optimized. Sample preparation for human plasma consisted of simple and rapid protein precipitation with a 1/5 ratio of sample to solvent using an acetonitrile solution containing the ISs. Moreover, acetonitrile was chosen because of its better performance in precipitating proteins compared to other solvents [50,51]. Additionally, in the DBS method, a methanol solution of the ISs was used to extract the analytes from the matrix. Other solvents such as acetonitrile, ethanol, and isopropanol were tested, but methanol showed superior extraction. For both matrices, the extract was diluted with MP-A to avoid solvent effects and approximate the composition of the initial mobile phase. Neither plasma extraction nor DBS extraction resulted in qualitative matrix effects at the retention time of the analytes, as preliminary evaluated using the post-column infusion method described by Bonfiglio et al. [52]. Post-column infusion was also used to evaluate whether ion suppression and/or enhancement phenomena were present between the analyte and the co-eluting IS, with no significant changes in signals detected (Appendix A).

### 3.2. Validation of the Method in Human Plasma

#### 3.2.1. Linearity

The heteroscedasticity of the calibration values was evaluated with an F test, and the statistical parameters calculated using the different w (1, 1/x, and 1/x^2^) are shown in Appendix A. Because the r values were very similar between the model with w = 1/x and the model with w = 1/x^2^ for all analytes, the choice was made based on the results for SS, AIC, and the sum of the absolute %RE parameters. For all analytes, the best performance was observed with w = 1/x^2^, although the differences between 1/x and 1/x^2^ as weighting factors were small. This result was also confirmed by the plot of %RE (Appendix A), where the narrowest horizontal band of %RE was observed when w = 1/x^2^ was applied. The reduction in variability is most evident at the lower concentrations of the calibration curves. Therefore, this weighting factor was used for all validation tests and clinical sample analysis for both plasma and DBS methods. The accuracy of calibrators was within 98–102%, and the highest CV% was 7%. The correlation coefficient r was 0.997 or better for all analytes, demonstrating good linearity within the concentration ranges considered (Appendix A).

#### 3.2.2. Accuracy and Precision

The A & P of the LLOQ and QCs for each analyte were within the recommended limits. As indicated in Table 2, the within-run accuracy ranged from 87 to 102%, and the highest CV% was 7%. Similarly, the between-run accuracy ranged from 96 to 102%, with the highest CV% recorded at 8%. These results demonstrate satisfactory accuracy and precision for the tested concentration levels and analytes.

#### 3.2.3. Recovery and Matrix Effect

Recovery was assessed by comparing the area ratio from the normal extraction with the area ratio from the post-extraction technique. The results from five replicates of the QCL, QCM, and QCH were within 98–103%, with the highest CV% value being 5%, demonstrating a high and reproducible extraction yield for all concentration levels considered (Table 3). As reported in Section 3.1, the post-column infusion test was used to do a qualitative evaluation of the matrix effect. This test showed that there was no ion suppression or enhancement at the retention time of the analytes and ISs (Appendix A). The matrix effect was evaluated by comparing the peak area obtained with the post-extraction technique with the peak area obtained with the pure solution of the analytes at the QCL and QCH concentration levels. The results showed that the highest CV% was 4%, making the influence of the single matrix negligible in the quantification of PARPis (Table 3).

#### 3.2.4. Selectivity and Sensitivity

Selectivity was assessed using seven replicates of blank samples prepared with the same matrices mentioned above for matrix effect. At the retention time of all analytes and internal standards (ISs), no interferences were detected. This confirms the selectivity of the technique, as there were no observed interferences from other substances in the samples. Sensitivity was evaluated using five replicates of LLOQ analyzed in three different analytical runs: accuracy ranged from 96 to 102%, the highest CV% was 8%, and the signal-to-noise ratio was always above 30. These results demonstrate satisfactory sensitivity for all analytes.

#### 3.2.5. Dilution Integrity and Stability

A dilution factor of five was validated to cover all the possible concentrations in patient samples: accuracy was within 103–108% and the highest CV% was 5%, demonstrating the integrity of the sample dilution (Appendix A). No dilution was necessary for the study population. This indicates that the range chosen was broad enough to cover all concentrations found so far in the present TDM study. Stability in plasma was evaluated and validated after 5 h at room temperature, after 194 days at −80 °C, and after 6 freeze–thaw cycles. Moreover, the stability of the final extract was demonstrated after 5 days under autosampler conditions (15 °C, dark). Accuracy ranged from 93 to 106%, and the highest CV% was 7% for all the conditions tested, indicating that no significant degradation occurred for all analytes (Table 4).

### 3.3. Validation of the Method in DBS

#### 3.3.1. Linearity

Linearity was evaluated using eight calibrators read in duplicate in three different analytical runs on three different days: accuracy was within 98–104% and the highest CV% was 4%; the correlation coefficient r was ≥0.999 for all analytes (Appendix A).

#### 3.3.2. Accuracy and Precision

A & P were determined similarly to human plasma, using calibrators and QCs prepared with an artificial Hct value of 36%. Hct 36% was chosen because it has been estimated as the mean Hct value in patient samples [39]. The within-run accuracy ranged from 94–105% and the highest CV% was 6%, while the between-run accuracy ranged from 95–104% and the highest CV% was 5% (Table 5).

#### 3.3.3. Hematocrit Effect

The validation of the Hct range (29–45%) was performed as described in Section 2.7.1. The accuracy for the Hct value of 29% fell within the range of 105–115% and the highest CV% recorded was 9%. Similarly, for the Hct value of 45%, the accuracy was within 96–109%, and the highest CV% observed was 9%. These results showed that calibrators with an Hct value of 36% were suitable for the analysis of samples from patients in the Hct range of 29–45% (Table 6). Moreover, the effect of Hct on A & P was also evaluated in the Hct range of 25–55%; however, the accuracy did not meet the regulatory requirements (Appendix A).

#### 3.3.4. Recovery and Matrix Effect

Recovery was within 56–57% with the highest CV% of 10% for niraparib, within 66–68% with the highest CV% of 8% for rucaparib, and within 92–94% with the highest CV% of 7% for olaparib (Table 7). As previously reported by Antunes et al. [31], we found that the least polar compound (i.e., olaparib) had the highest recovery. Instead, niraparib and rucaparib are polar weak bases and may have a higher affinity for the matrix (or paper) with respect to the extraction solvent, in our case, methanol. In addition, it was observed that recovery was slightly different for an Hct value of 29% than for an Hct value of 45%. This could be explained by the fact that blood with a lower Hct value has a lower viscosity and produces a larger blood spot than blood with a higher Hct value [53]. The matrix effect was evaluated similarly to human plasma, using six different donors and two different Hct values (29% and 45%), each at three concentration levels (QCL, QCM, and QCH) and in triplicate. This resulted in a total of 108 values per analyte. The results showed that the ISN-MF with the highest CV% was 8%, making the influence of the single matrix negligible in PARPis quantification with DBS (Table 7). It should be noted that the analytical run to determine the recovery and matrix effect was split over two different days due to the abundance of samples.

#### 3.3.5. Selectivity and Sensitivity

Selectivity was assessed using six replicates of blank samples prepared with the same matrices mentioned above for the matrix effect. At the retention time of all analytes and ISs, no interferences were detected. Sensitivity was evaluated using five replicates of LLOQ (Figure 2) analyzed in three different analytical runs: the accuracy ranged from 96 to 100%, the highest CV% was 5%, and the signal-to-noise ratio was always above 30. This shows satisfactory sensitivity despite the lower extraction recovery of rucaparib and niraparib.

#### 3.3.6. Carryover and Stability

As described in Section 3.1, there was no carryover. Finally, analyte stability was evaluated in DBS after 219 days under desiccator conditions, and after extraction, analytes were stable after 14 days from the first injection under autosampler conditions. Under both conditions, accuracy was in the range of 95–103%, and the highest CV% was 6% for rucaparib and olaparib. This indicates that no significant degradation occurred. Results for niraparib showed moderate degradation under both conditions: accuracy ranged from 91 to 96% (CV% ≤ 5%) after 14 days under autosampler conditions and from 90 to 92% (CV% ≤ 3%) after 219 days in the desiccator (Table 8).

### 3.4. Clinical Validation

To date, 42 patients have been enrolled in the present study, resulting in a total of 125 samples. However, due to sampling difficulties and an Hct value outside of the validated range, a total of 111 samples from 41 patients were considered for the clinical validation study. Specifically, 52 samples were collected from 16 patients for olaparib, 43 samples from 21 patients for niraparib, and 16 samples from 4 patients for rucaparib. All DBS samples were collected within 2 weeks of Hct determination, except for 4 samples containing olaparib. Hct levels ranged from 29 to 43%, and the time since the last dose was at least 9 h. The concentrations measured in the samples (C_DBS_ and C_pla_) collected from the patients are listed in the Appendix A.

The aim of this study was to compare the concentrations of PARPis in human plasma and in DBS to determine if both matrices could be used for sampling. In the simplest case, the drug is equally distributed between plasma and red blood cells and has the same concentration in both matrices; therefore, it is not necessary to convert C_DBS_ to C_pla_ [33]. However, in most cases, C_pla_ and C_DBS_ are different, and a CF must be used, as reported in several papers [31,32]. In our study, C_pla_ and C_DBS_ were also different, and it was found that the simple ratio of C_DBS_ to C_pla_ as CF was good enough for correlation. This result is advantageous because CF is a simple number and not a formula that contains parameters such as Hct, plasma fraction, and others [54]. In this last case, these parameters had to be known at the time of sampling, so a blood sample had to be analyzed on the same day as the DBS sampling. The CFs obtained were 1.427 for rucaparib, 1.440 for niraparib, and 0.718 for olaparib, as shown in Table 9.

The statistical analysis presented in this section highlights the agreement between EC_pla_ and C_pla_, whose values are reported in Appendix A. The results of the agreement are shown in Figure 3 and summarized in Table 9. The Passing–Bablok regression analysis between EC_pla_ and C_pla_ showed that there were no systematic and proportional errors for niraparib and olaparib because the intercept was close to zero and the slope was near 1. In particular, for niraparib, the intercept was −5 ng/mL and the slope was 1.02 (95% CI: −31–15 and 0.95–1.07, respectively). Similarly, for olaparib, the intercept was 1 ng/mL and the slope was 1.00 (95% CI: −34–30 and 0.97–1.03, respectively). In addition, the data showed good concordance, as confirmed by Lin’s CCC value of 0.960 for niraparib and 0.995 for olaparib. For rucaparib, the intercept was determined to be 92 ng/mL (95% CI: −284–385), while the slope was found to be 0.90 (95% CI: 0.73–1.17). The calculated Lin’s CCC value for rucaparib was 0.935. It is important to note, however, that the correlation analysis was based on a limited dataset consisting of only 16 samples. Therefore, the data should be regarded as preliminary, and further analyses are required to establish a more comprehensive understanding of the relationship between estimated and actual concentrations of rucaparib. In addition, the cumulative sum (cusum) linearity test showed no significant deviation from linearity as the p-value was >0.05 for all analytes.

The Bland–Altman plot showed no correlation patterns between the difference and the mean of EC_pla_ and C_pla_. This means that the difference between the two concentrations did not depend on the entity of measurement. Indeed, r_S_ was 0.0991 (*p* = 0.53) for niraparib and 0.0489 (*p* = 0.73) for olaparib. For niraparib, the mean bias was 0 ng/mL, and the measured mean concentration was 446 ng/mL. Moreover, almost all differences between EC_pla_ and C_pla_ were between −138 and 138 ng/mL, the lower and upper limits of agreement, respectively. For olaparib, the mean bias was −1 ng/mL, and the measured mean concentration was 1358 ng/mL. In this case, almost all differences between EC_pla_ and C_pla_ were between −205 and 203 ng/mL, the lower and upper limits of agreement, respectively. The results for rucaparib indicated a moderate correlation, probably due to the small sample size. However, the mean bias was −16 ng/mL and was very small compared with the measured mean concentration of 1523 ng/mL. Additionally, almost all differences between EC_pla_ and C_pla_ were between −432 and 400 ng/mL, the lower and upper limits of agreement, respectively. In addition, r_S_ was −0.2399 (*p* = 0.37), indicating a weak negative correlation between the differences among EC_pla_ and C_pla_ and the entity of the measurements.

Regarding the evaluation of %diff recommended by the EMA and FDA guidelines, our study observed a deviation greater than ±20% in 1 out of 16 (6.3%) samples for rucaparib, 5 out of 43 (11.6%) samples for niraparib, and 3 out of 52 (5.8%) samples for olaparib. These findings indicate a good correlation between EC_pla_ and C_pla_, highlighting the robustness of the developed method.

One limitation of this study is that knowledge of the patients’ Hct value is necessary within a time frame close to DBS sampling. This is because the performance of the method was validated within the Hct range of 29–45% despite using volumetric sampling. However, Hct determination on the day of sampling is not mandatory, which opens up the possibility of home sampling by patients themselves.

The CF, which is calculated as the ratio of the analyte C_DBS_ to the analyte C_pla_, provides a measure of the amount of analyte present in dried blood compared to plasma. This parameter can be associated with the blood-to-plasma ratio specific to a particular drug. The data obtained from the CF in this study could be compared with data from mass balance studies that used a radioactive agent to determine the compound’s preference for plasma or red blood cells. For olaparib, a blood-to-plasma ratio of 0.7 [4] was found in the literature, which agrees well with our CF value of 0.714. For rucaparib, previous studies reported CF values of 1.8 [6,10] and 1 [55], while our study obtained a CF value of 1.492, which falls within the range of these reported values. In the case of niraparib, previous literature reported blood-to-plasma ratios of 1.6 [11] and 1.7 [56]. Our study found a CF value of 1.476, which is in close agreement with the published data, further supporting the consistency of our findings. The agreement of the blood-to-plasma ratio between the literature and our result confirms the robustness of the present work. However, caution should be taken when preparing data because the blood-to-plasma ratio is known to depend on pH and temperature [57].

### 3.5. Incurred Sample Reanalysis

The EMA and FDA require that samples be reanalyzed in two different analytical runs to obtain ISR [36,37]. To evaluate the ISR, a total of three rucaparib samples, four niraparib samples, and four olaparib samples were analyzed in human plasma. Additionally, for the DBS matrix, four rucaparib samples, three niraparib samples, and six olaparib samples were analyzed. The differences in quantification between two separate analytical runs were less than 14% in all cases (Appendix A), demonstrating the reproducibility of the method and its applicability to study samples.

## 4. Conclusions

A new LC-MS/MS method for the determination of olaparib, rucaparib, and niraparib in human plasma and DBS was developed and validated for TDM purposes. The method proved to be suitable for C_trough_ quantification of PARPis, as all clinical samples were within the calibration range. Additionally, the sample preparation, based on a simple protein precipitation technique, was fast and straightforward. The validation results showed a high degree of robustness and reproducibility in both plasma and DBS matrices, which is necessary for routine analysis. For both plasma and DBS, the accuracy of the calibrators was within 98–104%, and the highest CV% was 7%, while good linearity was demonstrated by the correlation coefficient r of 0.997 or better for all analytes. Within-run accuracy was in the range of 87–105%, and the highest CV% was 7%, while between-run accuracy was in the range of 95–104%, and the highest CV% was 8% for both plasma and DBS. For DBS samples, the validated Hct range of 29–45% covers the great majority of the population studied, and the plasma method can be used if the Hct value of the sample is outside the above range. Plasma concentrations can be determined using a straightforward CF approach based on DBS measurements. The high degree of agreement between the calculated and the actual measured values was demonstrated in the clinical validation study conducted for olaparib and niraparib. However, for rucaparib, additional samples are required to provide a solid regression model. In conclusion, this study represents the first development and validation of a method for quantifying PARPis in the DBS matrix. This significant achievement opens up the concrete possibility of using DBS as an alternative and patient-friendly sampling method to promote the dissemination of TDM studies for PARPis.

## Figures and Tables

**Figure 1 pharmaceutics-15-01524-f001:**
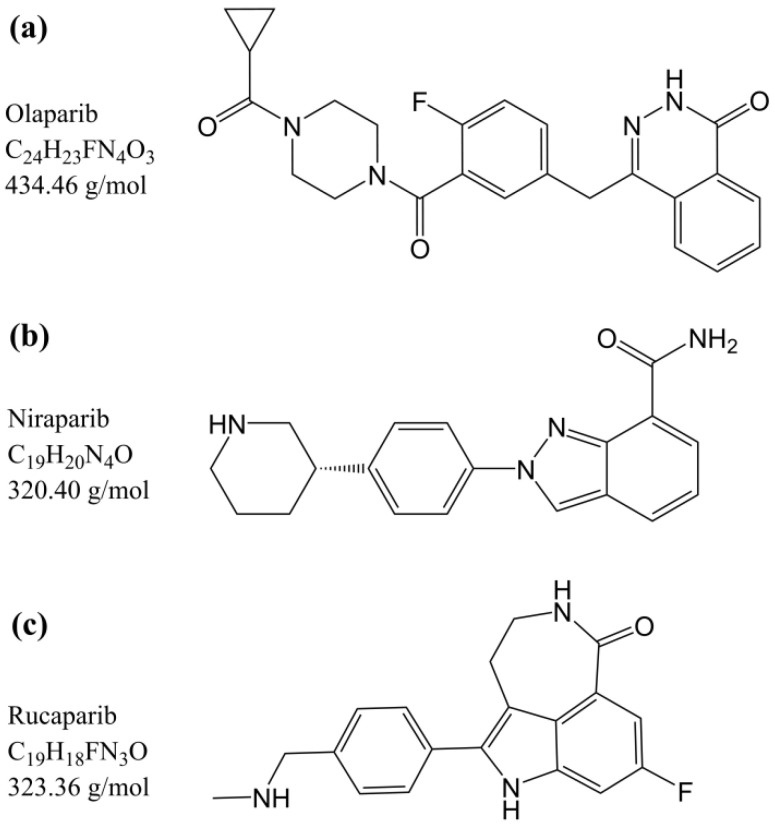
Chemical structures, molecular formula, and molecular weight of (**a**) olaparib, (**b**) niraparib, and (**c**) rucaparib.

**Figure 2 pharmaceutics-15-01524-f002:**
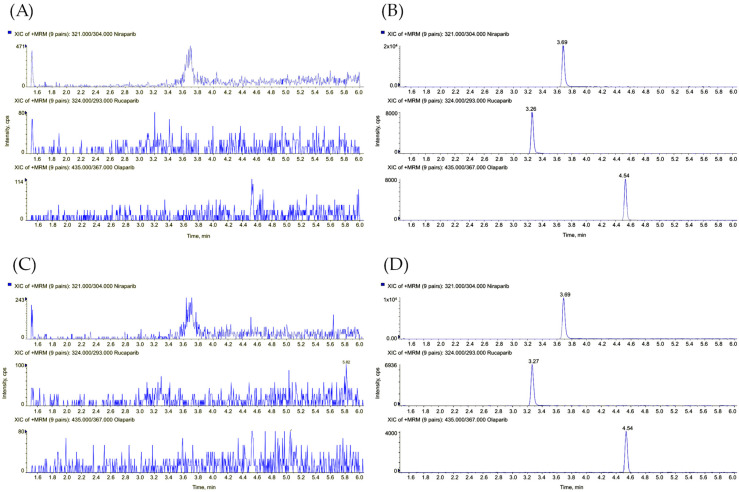
Liquid chromatography-tandem mass spectrometry (LC-MS/MS) chromatograms of (**A**) a blank human plasma sample, (**B**) rucaparib, niraparib, and olaparib spiked in human plasma at the lowest concentration levels (100 ng/mL, 60 ng/mL, and 140 ng/mL, respectively), (**C**) a blank DBS sample, and (**D**) rucaparib, niraparib, and olaparib in the DBS sample at the lowest concentration levels.

**Figure 3 pharmaceutics-15-01524-f003:**
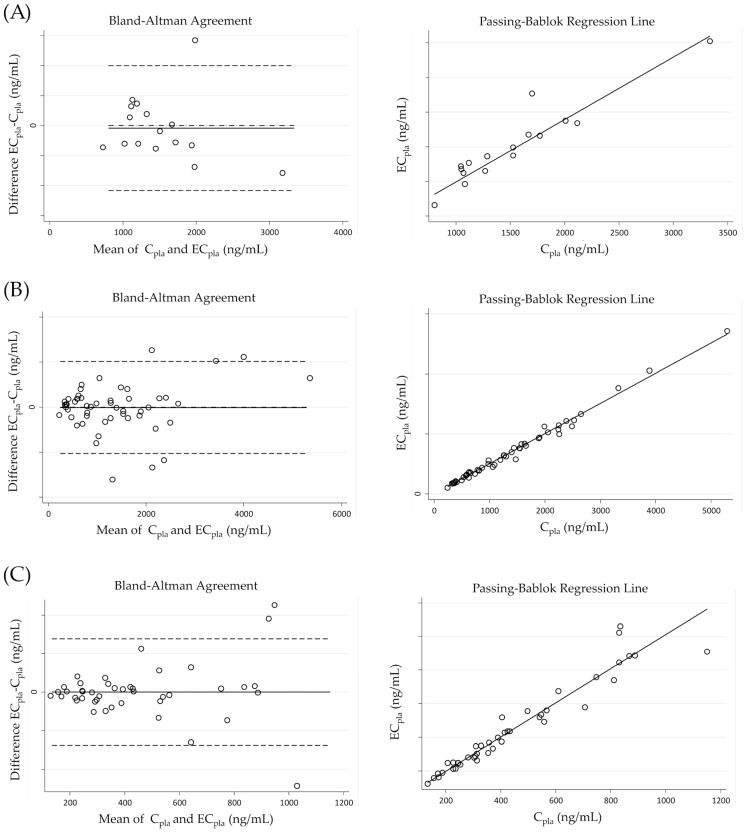
Bland–Altman agreement and Passing–Bablok regression line for (**A**) rucaparib, (**B**) olaparib and (**C**) niraparib. In Bland–Altman plots biases are represented by marked lines, while dashed lines represent either lower and upper limits of agreements (defined as ±1.96 SD—standard deviation—of the bias) and an ideal bias of 0 ng/mL. In Passing–Bablok plots regression lines are represented by marked lines.

**Table 1 pharmaceutics-15-01524-t001:** Analyte concentrations for calibrators and quality controls (QCs) in both human plasma and dried blood spot (DBS) matrices.

Concentration Level	Concentration (ng/mL)		
	Olaparib	Rucaparib	Niraparib
H	140	100	60
G	280	200	120
F	700	500	300
E	1400	1000	600
D	2800	2000	1200
C	4200	3000	1800
B	5600	4000	2400
A	7000	5000	3000
QC L	357	255	153
QC M	2380	1700	1020
QC H	5950	4250	2550

**Table 2 pharmaceutics-15-01524-t002:** Within- and between-run accuracy and precision (A & P) of the method in human plasma.

Analyte	Nominal Concentration (ng/mL)	Within-Run (*n* = 5)			Between-Run (*n* = 15)		
		Mean Concentration (ng/mL)	CV%	Acc%	Mean Concentration (ng/mL)	CV%	Acc%
rucaparib	100	87	6	87	96	8	96
	255	250	7	98	259	7	101
	1700	1708	7	100	1732	4	102
	4250	4190	5	99	4289	4	101
niraparib	60	61	2	101	60	3	101
	153	155	5	101	156	3	102
	1020	1043	2	102	1044	2	102
	2460	2442	2	96	2460	3	96
olaparib	140	139	3	99	143	4	102
	357	356	2	100	358	3	100
	2380	2361	3	99	2386	3	100
	5950	5802	3	98	5810	3	98

Acc%, percentage of accuracy; CV%, coefficient of variation; *n*, number of replicates.

**Table 3 pharmaceutics-15-01524-t003:** Recovery and IS-normalized matrix factor (ISN-MF) values of the method in human plasma.

Analyte	Nominal Concentration (ng/mL)	%Rec (*n* = 5)	CV%	ISN-MF (*n* = 7)	CV%
rucaparib	255	101	8	1.02	4
	1700	98	3	-	-
	4250	99	2	0.98	2
niraparib	153	98	2	1.06	2
	1020	99	1	-	-
	2460	98	3	0.97	2
olaparib	357	101	2	0.94	1
	2380	103	3	-	-
	5950	103	3	0.96	2

%Rec, percentage of recovery; CV%, coefficient of variation, ISN-MF, IS-normalized matrix factor; *n*, number of replicates.

**Table 4 pharmaceutics-15-01524-t004:** Stability data of rucaparib, niraparib, and olaparib in human plasma samples.

Analyte	Nominal Concentration (ng/mL)	5 h at RT (*n* = 3)			5 Days in Autosampler Conditions (Final Extract at 15 °C, Dark) (*n* = 3)			6 Freeze–Thaw Cycles (*n* = 3)			194 Days at −80 °C (*n* = 3)		
		Mean Concentration (ng/mL)	CV%	Acc%	Mean Concentration (ng/mL)	CV%	Acc%	Mean Concentration (ng/mL)	CV%	Acc%	Mean Concentration (ng/mL)	CV%	Acc%
rucaparib	255	254	5	100	259	6	102	270	4	106	251	6	98
	4250	4325	7	102	4321	1	102	4432	1	104	4273	3	101
niraparib	153	159	4	104	155	3	101	153	3	100	151	4	99
	2550	2477	2	97	2406	1	94	2458	2	96	2388	1	94
olaparib	357	347	3	97	335	1	94	377	2	106	357	1	100
	5950	5846	2	98	5549	1	93	6077	2	102	5860	1	98

Acc%, percentage of accuracy; CV%, coefficient of variation; *n*, number of replicates; RT, room temperature.

**Table 5 pharmaceutics-15-01524-t005:** Within- and between-run A & P of the method in DBS.

Analyte	Nominal Concentration (ng/mL)	Within-Run (*n* = 5)			Between-Run (*n* = 15)		
		Mean Concentration (ng/mL)	CV%	Acc%	Mean Concentration (ng/mL)	CV%	Acc%
rucaparib	100	95	6	95	96	5	96
	255	255	3	100	258	3	101
	1700	1784	3	105	1761	3	104
	4250	4402	4	104	4353	5	102
niraparib	60	60	3	101	59	4	98
	153	149	2	98	149	3	97
	1020	1026	3	101	1009	3	99
	2460	2483	3	97	2418	3	95
olaparib	140	132	2	94	140	5	100
	357	338	2	95	350	4	98
	2380	2373	2	100	2376	3	100
	5950	5859	1	98	5918	3	99

Acc%, percentage of accuracy; CV%, coefficient of variation; *n*, number of replicates.

**Table 6 pharmaceutics-15-01524-t006:** Hematocrit (Hct) effect on A & P of the method in DBS.

Analyte	Nominal Concentration (ng/mL)	29% Hct (*n* = 3)			45% Hct (*n* = 3)		
		Mean Concentration (ng/mL)	CV%	Acc%	Mean Concentration (ng/mL)	CV%	Acc%
rucaparib	100	108	9	108	102	2	102
	255	291	3	114	259	7	101
	1700	1917	3	113	1791	5	105
	4250	4797	4	113	4617	3	109
niraparib	60	63	4	105	57	6	96
	153	176	4	115	150	9	98
	1020	1143	2	112	1060	3	104
	2460	2894	4	113	2641	1	104
olaparib	140	156	6	112	150	3	107
	357	376	3	105	366	2	103
	2380	2497	3	105	2283	1	96
	5950	6473	5	109	5734	4	96

Acc%, percentage of accuracy; CV%, coefficient of variation; Hct, hematocrit; *n*, number of replicates.

**Table 7 pharmaceutics-15-01524-t007:** Recovery and ISN-MF values of the method in DBS.

Analyte	Nominal Concentration (ng/mL)	%Rec (*n* = 24)	CV%	ISN-MF (*n* = 24)	CV%
rucaparib	255	66	8	1.17	5
	1700	68	8	1.14	6
	4250	66	4	1.14	5
niraparib	153	56	10	1.36	7
	1020	56	6	1.38	8
	2460	57	5	1.35	8
olaparib	357	93	7	1.06	7
	2380	94	6	1.03	8
	5950	92	6	1.02	7

%Rec, percentage of recovery; CV%, coefficient of variation, ISN-MF, IS-normalized matrix factor; *n*, number of replicates.

**Table 8 pharmaceutics-15-01524-t008:** Stability data of rucaparib, niraparib, and olaparib in DBS samples.

Analyte	Nominal Concentration (ng/mL)	14 Days in Autosampler Conditions (Final Extract at 15 °C, Dark) (*n* = 3)			219 Days in Desiccator Conditions (20 °C, Humidity < 35%) (*n* = 3)		
		Mean Concentration (ng/mL)	CV%	Acc%	Mean Concentration (ng/mL)	CV%	Acc%
rucaparib	255	257	2	101	252	1	99
	4250	4336	5	102	4018	4	95
niraparib	153	147	5	96	141	1	92
	2550	2330	1	91	2301	3	90
olaparib	357	356	6	100	368	1	103
	5950	5964	3	100	6104	1	103

Acc%, percentage of accuracy; CV%, coefficient of variation; *n*, number of replicates.

**Table 9 pharmaceutics-15-01524-t009:** Agreement of plasma and DBS measurements: results of Passing–Bablok regression and Bland–Altman analysis.

Analyte	CF	Passing–Bablok Regression			Lin’s CCC	Bland–Altman Analysis		
		Slope (95% CI)	Intercept (95% CI) *	Cusum Test		Bias (95% CI) *	SD *	Spearman Correlation
rucaparib	1.427	0.90 (0.73–1.17)	92 (−284–385)	*p* > 0.05	0.935	−16 (−129–97)	212	r_S_ = −0.2399 (*p* = 0.37)
niraparib	1.440	1.02 (0.95–1.07)	−5 (−31–15)	*p* > 0.05	0.960	0 (−22–22)	71	r_S_ = −0.0991 (*p* = 0.53)
olaparib	0.718	1.00 (0.97–1.03)	1 (−34–30)	*p* > 0.05	0.995	−1 (−30–28)	104	r_S_ = 0.0489 (*p* = 0.73)

* values reported in ng/mL; CF, conversion factor; CI, confidence interval; Lin’s CCC, Lin’s concordance and correlation coefficient; SD, standard deviation; Cusum: cumulative sum.

## Data Availability

Data are contained within the article or Appendix A.

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
