# Peer review of "LC-MS/MS Method for the Quantification of PARP Inhibitors Olaparib, Rucaparib and Niraparib in Human Plasma and Dried Blood Spot: Development, Validation and Clinical Validation for Therapeutic Drug Monitoring"

_pharmaceutics, 2023, doi:10.3390/pharmaceutics15051524_

Round 1
Reviewer 1 Report (New Reviewer)
The manuscript entitled “LC-MS/MS Method for the Quantification of PARP inhibitors Olaparib, Rucaparib and Niraparib in Human Plasma and Dried Blood Spot: Development, Validation and Clinical Validation for Therapeutic Drug Monitoring” describes development and validation of a liquid chromatography-tandem mass spectrometric (LC-MS/MS) method for olaparib, rucaparib and niraparib quantification in both human plasma and dried blood spot matrices, and to evaluate their correlation. However, the presented article could be published in Pharmaceutics after revision for the following.
1. Selection of LQC, MQC and HQC should be justified.
2. Table 1: it is better to transfer to supplementary data.
3. Table S6, S7 and S9 could be merged into one table and placed in the original article.
4. Section 2.6.2. was missed.
5. Introduction part should be enriched with a sufficient background as well as a justification for the current research purpose.
English should be polished throughout the text.
Author Response
Please see the attachment

Reviewer 2 Report (New Reviewer)
pharmaceutics-2379605
LC-MS/MS Method for the Quantification of PARP inhibitors Olaparib, Rucaparib and Niraparib in Human Plasma and Dried Blood Spot: Development, Validation and Clinical Validation for Therapeutic Drug Monitoring
The manuscript by Canil et al. described the development and validation of an LC-MS/MS method for the analysis of olaparib, rucaparib, and niraparib (PARP inhibitors) in both human plasma and dried blood spot matrices. The PARP inhibitors in DBS matrices analyzed by the devised method showed a correlation to their plasma analog and can be used for TDM routine analysis. Overall, the authors provided sufficient data to support the development and validation of the method. However, there are some minor points as follows. Please consider them to improve the manuscript.
1. The tables should be in the same style.
2. In tables, it is better to add the number of replication (n) for clarification.
3. The m/z transitions of PARP inhibitors and the IS should be interpreted and verified based on their structure.
None
Author Response
Please see the attachment

Reviewer 3 Report (New Reviewer)
This study described the LC-MS/MS Method for the quantification of three PARP inhibitors Olaparib, Rucaparib and Niraparib in Human Plasma and Dried Blood Spot. The study was clinical validated in therapeutic drug monitoring. Authors claim novelty as this is first method for analysis of target analytes in DBS. The manuscript fits within the scope of the journal and is also interesting. However, the following issues need to be addressed properly for further improvement of the manuscript. These are:
1. In introduction section authors mentioned that previous study with same analytes was reported with UHPLC with 2µm column and which is not commonly available in bioanalytical lab. I think this sentence author need to omit as now most of the lab is adopted for UHPLC based analysis due to lots of advantages. Moreover, author should need to provide the comparative advantaged of this method compared to previous method in term of sensitivity, mode of sample preparation and analysis time.
2. Since sensitivity of this method is poor in compression to previous methods, author should ensure the assay suitability for covering the trough samples concentration level based on recommend doses in patients.
3. The languages of the manuscript are very poor. In addition, there are several formatting and syntex issues in the manuscript which need to be edited by native language professional.
4. In DBS validation study, the evaluation of effect of blood spot volume on analytes concentration is missing.
5. In Figure 2, authors need to shift the the representative chromatgorams of analyes in DBS from supplementary materials and vice versa as this method is focused on DBS. In addition, blank chromatograms in DBS is also need to provide here.
6. IS-normalized matrix effects result in DBS is found to higher than the expected at some concentration levels. Is it can be considered as negligible? Kindly justify
7. The mass spectra of precursor to product ion fragmentation is missing in the manuscript.
8. It is suggested to re-organize the conclusion section much better. The conclusion is too much and lacks some basic components of validation results. Kindly summarize and be re-written in a well-structured manner
The languages of the manuscript are very poor. In addition, there are several formatting and syntex issues in the manuscript which need to be edited by native language professional.
Round 2
Reviewer 3 Report (New Reviewer)
Authors addressed the comments properly and now can be consider for publication.
This manuscript is a resubmission of an earlier submission. The following is a list of the peer review reports and author responses from that submission.
Round 1
Reviewer 1 Report
General
The manuscript submitted by Canil et al. describes a well designed LC-MS/MS method for quantification of olaparib, rucaparib and niraparib in human plasma and Dried Blood Spot (DBS) samples. Although there are already several LC-MS methods described for these compounds, the presented method and its application for DBS samples is relevant. Validation of the method according to EMA and FDA guidelines as well as comparison of results determined for human plasma and Dried Blood Spot (DBS) samples represents the main part of the manuscript. A wealth of original data is presented in SI enabling a detailed vision on the results but making it also a bit tedious to go through all details. It is well structured and properly written. The results described regarding validation are convincing. Investigation of plasma results vs DBS is plausible, and the same is mostly true for the calculation of plasma concentrations based on the DBS data. There are, however, some concerns remaining.
Concerns
LC-MS-method
· The authors make use of isotopically labelled IS for all analytes, enabling a high degree of reliability for quantification in the investigated matrices. Therefore, some of the described efforts e.g. post column infusion to study ME or IS normalised MF (please explain determination/calculation of IS in Experimental section or SI) would be dispensable.
· Some of the method development steps are a bit uncommon, ambiguously described or a bit questionable: (I) Optimisation of analyte solutions in pure methanol for tuning of compound dependent parameters? (II) In figure 2 rucaparib is the compound with the lowest intensity at the LLOQ, therefore intensity of source parameters should have been optimised for this compound and not for niraparib. (III) For optimisation of source parameters FIA of matrix samples seems to more efficient. (IV) Analyte and IS are distinctly different in their m/z, therefore a crosstalk cannot occur, even if the IS fragment has the same m/z as the analyte (this is at least true for the Sciex QqQ-MS systems).
· In comparison to a linear regression without weighting, application of 1/x2 weighting does not reduce the absolute sum of residues (RSS). Typically, the sum of the absolute values of the relative error (ΣRE) is reduced (the problem occurs also in SI).
· Section 2.5 “while the ESI source parameters were optimised also with the LC system pumping a 50:50 mixture of the mobile phases” → different mobile phases? → presumably the two components of the mobile phase, please clarify
Validation
· With respect to recovery results (section 3.3) speculations should be reduced, if there is no evidence from experiments
Clinical application and correlation plasma DBS
· For rucaparib the number sample/patients is very low, maybe too low to demonstrate a meaningful correlation. This should be mentioned in the conclusion. The message in the abstract addressing this issue should be expressed more precisely (i.e. it is not enough do continue with correlation models, necessary are additional samples/results for a meaningful comparison/correlation).
· In fig 3 A/C Passing-Bablok Regression the x-axis should start at 0 (as in B).
· Section 3.4 “The ultimate goal of this study was to compare the concentration of PARPis in human plasma and in DBS matrices, in order to evaluate if either of them could be used for sample collection” There, should be literature demonstrating this for plasma.
· The way Spearman correlation is done should be described in the experimental section. The obtained r values are rather low. This would mean that ECpla calculation based on DBS is not very reliable.
SI
· The context of the experimental conditions of Table S2 and Fig S1 is not clear. Description of the conditions to determine dwell volume and ECV?
· Issue of weighting – see comments before.
Minor points
· Some typos, e.g. SI weigthing
· Positive ion mode → better positive ionisation mode
Reviewer 2 Report
Manuscript: LC-MS/MS method for the quantification of PARP inhibitors olaparib, rucaparib and niraparib in human plasma and Dried Blood Spot (DBS) matrices for therapeutic drug monitoring
The manuscript is clearly and concisely written, follows the latest recommendations and has significant clinical application. These are my general comments:
1. The manuscript is actually an analytical validation of the LC-MS/MS method for the quantification of PARP inhibitors olaparib, rucaparib and niraparib in human plasma and Dried Blood Spot (DBS) matrices for future clinical application, so it would be good if that was clear from the title.
2. The concentrations of calibrators and controls are not listed in the text but in the supplementary materials, so they should be listed.
3. The validation was done according to the valid guidelines, but the results are presented in the text and quite unclear, so they should be presented concisely and clearly in a table according to concentrations and sample number.
4. Method was developed for measuring PARP inhibitors in plasma and dried blood spot and expected values ​​or reference intervals are not mentioned. The clinical application of the method for therapeutic monitoring needs therapeutic and toxic concentrations if they are defined and also needs validation of therapeutic intervals. Can you comment this in the manuscript.
Reviewer 3 Report
This manuscript is for an LC-MS/MS method development for three PARP inhibitors in DBS and in human plasma for TDM purpose. The topic looks interesting, and I have a few questions to ask to the authors as follows:
1. Generally abbreviation is not used in the title. It would be nice to give full name instead of short one (LC-MS/MS).
2. It seems unclear how much TDM monitoring would be required in this PARP class drugs in real patient's routine life. Authors mentioned Niraparib as an example but the human PK for these PARP compounds are already well known and therefore the Ctrough would be easily predicted after administration without TDM using DBS, I guess. In addition, DBS is a just sample collection tool not an instrument that can provide the estimated conc in plasma to patients at home. In other words, patients would still need to bring the DBS samples they prepared at home to the clinics and get these samples measured by LC-MS/MS in the hospital which takes a couple of days (or sometimes more than two weeks) at least to know the concentrations (Ctrough) and this might not be realistic from TDM perspectives. Nevertheless, this manuscript is still valuable from the comparison perspectives between two sample collection methods ( DBS vs. human plasma ) and I hope the authors would be able to modify the introduction to reflect this as well.
3. In the section 3.4, some samples were not considered or DBS analysis due to Hct range was outside of the validated range. Is the Hct critical for the accuracy/precision of DBS? It would be ok as long as the PARP drugs could be extracted efficiently from DBS regardless of the Hct vaues. If authors can add DBS data with outside Hct samples in the manuscript, it would be helpful for readers to understand the importance of Hct range in the DBS analysis. Similar description is also presented in the section 3.4, page 10 (One limitation of this study is~).
